# Comparing the Clinical Efficacy of Coil Embolization in GDA Stump versus Common Hepatic Artery in Postoperative Hemorrhage after Pancreatoduodenectomy

**DOI:** 10.3390/jpm13020264

**Published:** 2023-01-31

**Authors:** Chia-Chien Wu, Huan-Wu Chen, Ker-En Lee, Yon-Cheong Wong, Yi-Kang Ku

**Affiliations:** 1Department of Medical Imaging and Intervention, New Taipei Municipal Tu Cheng Hospital, Chang Gung Medical Foundation, New Taipei City 236, Taiwan; 2Division of Emergency and Critical Care Radiology, Department of Medical Imaging & Intervention, Chang Gung Memorial Hospital, Taoyuan 333, Taiwan; 3College of Medicine, Chang Gung University, Taoyuan 330, Taiwan

**Keywords:** postoperative hemorrhage, pancreatoduodenectomy, GDA stump, hepatic artery sacrifice

## Abstract

Background: Hemorrhage after pancreaticoduodenectomy is an uncommon but fatal complication. In this retrospective study, the different treatment modalities and outcomes for treating post-pancreaticoduodenectomy hemorrhage are analyzed. Methods: Our hospital imaging database was queried to identify patients who had undergone pancreaticoduodenectomy during the period of 2004–2019. The patients were retrospectively split into three groups, according to their treatment: conservative treatment without embolization (group A: A1, negative angiography; A2, positive angiography), hepatic artery sacrifice/embolization (group B: B1, complete; B2, incomplete), and gastroduodenal artery (GDA) stump embolization (group C). Results: There were 24 patients who received angiography or transarterial embolization (TAE) treatment 37 times (cases). In group A, high re-bleeding rates (60%, 6/10 cases) were observed, with 50% (4/8 cases) for subgroup A1 and 100% (2/2 cases) for subgroup A2. In group B, the re-bleeding rates were lowest (21.1%, 4/19 cases) with 0% (0/16 cases) for subgroup B1 and 100% (4/4 cases) for subgroup B2. The rate of post-TAE complications (such as hepatic failure, infarct, and/or abscess) in group B was not low (35.3%, 6/16 patients), especially in patients with underlying liver disease, such as liver cirrhosis and post-hepatectomy (100% (3/3 patients), vs. 23.1% (3/13 patients); *p* = 0.036, *p* < 0.05). The highest rate of re-bleeding (62.5%, 5/8 cases) was observed for group C. There was a significant difference in the re-bleeding rates of subgroup B1 and group C (*p* = 0.00017). The more iterations of angiography, the higher the mortality rate (18.2% (2/11 patients), <3 times vs. 60% (3/5 patients), ≥3 times; *p* = 0.245). Conclusions: The complete sacrifice of the hepatic artery is an effective first-line treatment for pseudoaneurysm or for the rupture of the GDA stump after pancreaticoduodenectomy. Hepatic complications are not uncommon and are highly associated with underlying liver disease. Conservative treatment, the selective embolization of the GDA stump, and incomplete hepatic artery embolization do not provide enduring treatment effects.

## 1. Introduction

Pancreatoduodenectomy is one of several operation methods for treating diseases of the pancreatic head and periampullary region (such as the second portion of the duodenum and the distal CBD). Thanks to advances in operative techniques, the complication rate has been reduced, especially in high-volume centers. However, delayed hemorrhage (beyond 24 h after surgery) is still one of the most harmful and fatal complications. It often occurs 1–4 weeks after the pancreatoduodenectomy. The most common internal bleeding site is the gastroduodenal artery (GDA) stump, with or without the presence of a pseudoaneurysm [1]. Bleeding is usually considered to be correlated with local inflammation and corrosion, due to pancreatic anastomosis leakage [2,3,4]. To achieve hemostasis and improved outcomes, it is suggested that urgent angiography be performed to identify the location of the internal bleeding, with subsequent embolization by radiological intervention [5]. Several previous studies have also supported the role of transarterial embolization (TAE) as a first-line treatment in delayed massive internal hemorrhage after pancreaticoduodenectomy [6,7,8,9].

However, there is controversy regarding the optimal way to carry out embolization. Some studies have shown that sacrificing the common/proper hepatic artery, both proximal and distal to the GDA stump, results in the best hemostatic effect [6,10]; in contrast, other studies have shown an equal outcome for the selective embolization of GDA stumps with/without pseudoaneurysm formation [11,12]. Therefore, this study aims to evaluate the safety and efficacy of conservative treatment and three embolization techniques, in order to establish a therapeutic strategy for the treatment of delayed massive hemorrhage after pancreaticoduodenectomy.

## 2. Materials and Methods

### 2.1. Patients

This is a retrospective and single-center study. We retrospectively reviewed the records of patients who underwent surgical resection between January 2004 and December 2019 at the Chang Gung Memorial Hospital (CGMH), Taiwan. Post-pancreatoduodenectomy hemorrhage data were collected from the CGMH imaging dataset. This protocol was approved by the CGMH Institutional Review Board and the Ethics Committee. The Ethics Committee waived the requirement for informed consent for this study. All methods in this study were carried out in accordance with the institutional guidelines and regulations of the Chang Gung Medical Foundation. The inclusion criteria were as follows: (a) patients received pancreatoduodenectomy or any periampullary region operation, with GDA ligation; (b) patients had clinical evidence of internal bleeding, with or without a positive finding on the computed tomography (CT) scan with contrast and/or the CTA (CT angiography) scan; (c) patients received angiography. This study excluded (a) patients without GDA ligation, and (b) cases where the patients’ bleeding site was not from the GDA stump. 

We used an electronic search method to identify the medical records of patients treated at our hospital. The patients’ medical records and radiological images were retrospectively reviewed in the electronic database of our institution by assessing preoperative characteristics, postoperative hemorrhage, additional interventions, mortality, and postoperative complications (such as hepatic failure, abscess, or infarct).

There is a high prevalence of viral hepatitis in Taiwan; viral hepatitis induces liver cirrhosis (a calculated annual incidence of 2.4%) [13,14], which subsequently causes portal hypertension and affects the entire liver perfusion. Because liver perfusion via the hepatic artery is sacrificed after TAE and may cause severe post-TAE complications, such as liver infarction, secondary liver abscesses, and hepatic failure, we included underlying liver disease (such as liver cirrhosis and any liver-related surgery) to evaluate their association with post-TAE complications.

All the patients’ vital signs were closely monitored, some through laboratory blood tests (especially in relation to liver function), in the intensive care unit after the TAE procedure. If patients showed any clinical signs or symptoms suggesting re-bleeding, repeated angiography with subsequent embolization was performed. The follow-up duration was measured from the time of surgery until death or after discharge, until 30 June 2022 (more than 6 months). No patient was lost to the follow-up.

### 2.2. TAE Procedure

An abdominal CT scan with contrast and/or a CT angiography (CTA) scan was then obtained. If signs of active bleeding (contrast pooling and/or extravasation) or pseudoaneurysm were observed on the CT scan, angiography was suggested. If there was no active bleeding sign but clinical suspicion remained high, with or without unstable vital signs, angiography was also indicated. The right or left common femoral artery was accessed using a puncture needle, which was then exchanged for a size 5 French sheath via the Seldinger technique. An angiogram of the celiac trunk, the superior mesenteric artery, and/or the common hepatic artery was obtained with an angiographic catheter for evaluation of the bleeding site. Positive angiography findings were defined as contrast extravasation/pooling or pseudoaneurysm. After evaluation of the GDA stump, with or without pseudoaneurysm, conservative treatment or TAE was performed. The patients were retrospectively split into three groups, according to their treatment: conservative treatment without embolization (group A), hepatic artery sacrifice/embolization (group B), and GDA stump embolization (group C).

In group A, conservative treatment without TAE was performed under close observation in the intensive care units. This group was divided into two subgroups: subgroup A1, for those who had negative angiography findings, and subgroup A2, for those who had hemodynamically stable vital signs, with positive contrast extravasation and/or pseudoaneurysm.

Two embolization techniques were performed: hepatic artery sacrifice/embolization (group B) and GDA stump embolization (group C). In group B, a microcatheter was advanced through the hepatic artery, distal to the GDA stump. Then, micro-coils were deployed to achieve the embolization of the proper/common hepatic artery, proximal to the GDA stump. After embolization, a complete celiac angiogram was performed to confirm the effects of TAE, such as sacrifice/complete (subgroup B1) or incomplete (subgroup B2) occlusion/embolization of the hepatic artery. The sacrifice of the hepatic artery involves embolism opacification in the hepatic artery, without patent blood flow; incomplete embolization involves embolism opacification in the hepatic artery, with residual partial hepatic artery flow. In group C, selective embolization of the GDA stump and/or pseudoaneurysm was carried out to preserve the hepatic arterial flow. Micro-coils were deployed to achieve the complete occlusion of the GDA stump.

### 2.3. Analysis

For the purposes of this study, post-pancreaticoduodenectomy hemorrhage was defined as the presence of clinical hypovolemia with or without fresh blood in the surgical drains or nasogastric tubes, hematemesis, and/or melena. A procedure was defined as a success when there was an absence of bleeding for 24 h, and re-bleeding was defined as a hemorrhage that occurred after initial hemostasis, more than 24 hours after the procedure. Complications were defined as events that required therapy, the unplanned deterioration of hepatic function, liver infarction with or without a liver abscess, and multi-organ failure. One case indicated a particular time of angiography, and each angiography was counted as a different case. Therefore, one patient may be seen to represent multiple cases. A procedure was defined as a success when there was an absence of bleeding for 24 h.

Because a smaller sample size may induce more bias, we only compared the rebleeding rate and complication rate in the larger groups. Baseline characteristics of the two different groups or subgroups were compared using Fisher’s exact tests for categorical variables, as appropriate, and these were presented as absolute numbers and percentages. Fisher’s exact test was used to determine the statistical differences between the different types of TAE, due to the small sample size. Values of *p* < 0.05 were considered to indicate a statistically significant difference. All statistical analyses were performed using IBM SPSS Statistics, version 20.0 (IBM Corp., Armonk, NY, USA).

## 3. Results

### 3.1. General Data Analysis

There were 24 patients (18 men and 6 women; mean age: 67.2 years, range: 37–81) who collectively received angiography 41 times, with or without subsequent transcatheter arterial embolization (TAE) to treat post-operation internal hemorrhage. Most of the preoperative indications for surgery were malignancy neoplasms, such as ampulla of Vater cancer (N = 11), choledochal cancer (N = 6), pancreatic cancer (N = 5), and duodenal adenocarcinoma (N = 1). In only one case was the preoperative indication a benign ampulla of Vater neoplasm, and pre-and post-operation pathology showed intestinal metaplasia of the bile duct, with surgery indicated due to the clinical suspicion of ampulla of Vater cancer and high-grade CBD obstruction with hyperbilirubinemia (direct/total bilirubin, 2.7/3.1 mg/dL) (Figure 1). All the described patients underwent elective operative procedures, including a pylorus-preserving pancreaticoduodenectomy (N = 9) and Whipple pancreaticoduodenectomy (N = 14), as well as cholecystoenterostomy and gastrojejunostomy, with partial pancreatectomy and GDA ligation (N = 1). The onset of internal bleeding occurred postoperatively from days 5 to 59 (median = 21.6, SD = 14.5).

### 3.2. Procedure Analysis

For the 1st time angiography, the rate of positive finding was 83.3% (20/24) with a sensitivity of 100%, specificity of 100%. These 24 patients with 37 angiographies were di-vided into three groups according to their treatment plan: conservative treatment without embolization (group A, n = 10; A1, n = 8, A2, n = 2), hepatic artery sacrifice/embolization (group B, n = 19; B1, n = 15, B2, n = 4), and GDA stump embolization (group C, n = 8). There was no obvious difference in the distribution due to the TAE type or year when the procedure was carried out (Figure 2). Figure 3 is a flowchart of all 24 patients enrolling procedure.

In group A, there were 10 cases without embolization during angiography; the re-bleeding rate was 60% (6/10). In subgroup A1, there were eight cases that had negative angiographic findings; TAE was not performed at that time. The success rate in subgroup A1 was 50% (4/8). Four cases were of re-bleeding, and all were treated with subsequent TAE. In subgroup A2, these two cases had positive angiographic findings, but their families declined TAE treatment at that time. The reason for this refusal, according to the admission chart records, was that the patients’ families hesitated due to the risk of post-TAE complications and wanted the patients to receive more conservative treatment first. Re-bleeding occurred in both cases; one patient was treated with complete TAE of the hepatic artery, while the other was discharged against medical advice. Both patients died of multiple organ failure due to profound shock. Thus, the re-bleeding rate of patients with positive angiography findings who opted for a more conservative treatment in subgroup A2 was 100% (2/2).

In group B, embolization of the hepatic artery was performed 19 times. The re-bleeding rate was 21.1% (4/19). In subgroup B1, there were 15 cases that involved hepatic artery sacrifice, with the patients having completely interrupted intrahepatic arterial flow through the common hepatic artery (Figure 4). The procedure success rate was 100%, with no occurrence of re-bleeding. In subgroup B2, four patients received partially interrupted hepatic artery embolization (Figure 5), and the procedure success rate was 100%, but the re-bleeding rate was also 100%. Three patients underwent another session of TAE, with the complete trapping of the hepatic artery, and immediate hemostasis was achieved; one patient died due to shock-induced multiple organ failure. There was a significant difference in re-bleeding rates between group B1 and group B2 (0% vs. 100%; *p* = 0.0003 < 0.05).

In group C, the GDA stump was completely embolized with the patent hepatic artery (Figure 6). There were five patients in this group who underwent angiography eight times, one underwent embolization of the GDA stump two times, and another underwent embolization of the GDA stump three times. The two patients who underwent GDA stump embolization experienced re-bleeding, and the patients underwent a session of TAE with hepatic artery sacrifice. The procedure success rate was 100%, although the re-bleeding rate was 62.5% (5/8).

Although there was no significant difference in the re-bleeding rates of groups B and C (21.1% vs. 62.5%; *p* = 0.072), there was a significant difference in the re-bleeding rates of subgroup B1 and group C (0% vs. 62.5%; *p* = 0.002 < 0.05).

The rate of complications for group B was 37.5% (6/16 patients), with six patients having experienced hepatic failure due to liver infarction, two of whom had liver abscesses. Three of these six patients had underlying liver disease and three did not. The patients with underlying liver disease had a higher risk of post-TAE complications (100% (3/3) vs. 23.1% (3/13); Fisher’s exact test statistic = 0.036, *p* < 0.05). Five of the patients in group B died; 18.2% (2/11 patients) of the patients received angiography less than three times, and 60% (3/5 patients) of the patients received angiography more than or equal to three times (*p* = 0.245). The sample size of the other groups was too small to allow further analysis.

## 4. Discussion

Delayed internal hemorrhage after pancreaticoduodenectomy is a rare complication with a very high mortality rate [15,16]. Bleeding is a result of vascular erosion related to bile and pancreatic juice, which is rich in proteolytic enzymes from pancreatic duct leakage and/or local infection [16,17,18]. Although postoperative internal hemorrhage is most often observed within 4 weeks after an operation, there were five patients in this study where bleeding occurred more than 4 weeks after the operation. The surgeon should keep in mind that delayed hemorrhage may occur up to 2 months after an operation and that bleeding must not be taken lightly.

If there is clinical evidence of internal bleeding, urgent angiography (to identify the location of the bleeding) with subsequent embolization by radiological intervention is the first approach toward achieving hemostasis. Some bleeding may be caused by mild damage to the arteries and/or veins; this can stop spontaneously, so the initial angiography may indicate negative findings. Because of the high sensitivity (100%) and specificity (100%) of negative findings for initial angiography in our study, as well as due to the low false negative rate (13%) and low re-bleeding rate and mortality (also supported by the study conducted by Kalva, S.P., et al. [19]), conservative treatment with close observation and without timely intervention is acceptable for these patients.

However, the vessel may easily spasm due to shock status, and the initial angiography may show negative findings. Once these patients present clinical evidence of re-bleeding, repeated angiography is indicated as a matter of urgency. The patients that underwent angiography more times in this study (which means more incidents of re-bleeding) also had a higher mortality rate (60%, more than or equal to three times, vs. 18.2%, less than three times). This means that patients with re-bleeding events are at greater risk of death. Therefore, immediate F/U angiography with hemostasis is the best approach for patients with re-bleeding [20]. We recommend that such patients receive TAE at the second angiography, regardless of whether the second angiography is positive or negative. Prophylactic hepatic artery sacrifice in patients could be of help in the treatment of suspicious bleeding sites (acute hematoma, contrast extravasation, or pooling) that have been revealed in previous CT or CTA scans [15]. For patients in an unstable situation, even if there are no available previous CTA scans and/or negative angiographic findings, prophylactic TAE of the complete hepatic artery may help because the GDA stump is the most common site of bleeding in cases of post-pancreaticoduodenectomy hemorrhage.

The re-bleeding rate was 0% for subgroup B1, 100% for subgroup B2, and 62.5% for group C; therefore, embolization of the GDA stump/pseudoaneurysm and incomplete hepatic artery embolization are not enduring solutions [6,21,22], especially in the case of post-TAE re-bleeding. Saebeom Hur et al. reported that “re-bleeding can occur not only at the GDA stump again, but also along any other portions of the extrahepatic segments of the hepatic artery” [6]. This is also in accordance with the findings of our study and explains the importance of and necessity for hepatic artery sacrifice. Hence, hepatic artery scarification as a method of TAE is preferred over the partial occlusion of the hepatic artery (B2) and GDA stump occlusion (C).

Most patients and their families hesitate in agreeing to hepatic artery scarification, perceiving that it will cause hepatic failure or infarction. This results in delayed treatment, and they may miss the best opportunity for hemostasis. The liver has a dual blood supply, with blood coming from the hepatic artery (30%) and the portal vein (70%). Therefore, the rates of complications and mortality are very low after hepatic artery sacrifice if the portal vein is patent [10]. Yekebas et al. reported successful blind coiling in four out of five patients, whereas only one patient required re-laparotomy [23]. In our patients, the prevalence of liver infarction and liver function impairment after TAE is not uncommon. The reason is that there is a high prevalence of hepatitis B-induced liver cirrhosis in our country, and patients also received post-hepatectomy; both of these issues decrease the blood supply from the portal vein. Therefore, if the patient has an underlying hepatic disease, such as liver cirrhosis, or has had prior hepatic surgery, the risk of post-TAE complications increases significantly (100% vs. 23.1%; *p* = 0.036 < 0.05). Choi, W.S., et al. reported similar findings [24]. Therefore, we should be more prudent in choosing the TAE treatment for patients with underlying hepatic diseases. One patient in our study underwent hepatic artery stenting due to underlying hepatic disease (Figure 7). Stenting may be an effective way to cease bleeding and preserve hepatic arterial flow, in order to decrease post-TAE complications. However, one reason for this approach being less widely used in Taiwan is that it is expensive and such operations were not covered by National Health Insurance prior to 2015. There are also other factors that should be matched among the criteria for stenting before coming to a conclusion. These factors include the tortuosity and diameter of the hepatic artery, how critical is the condition of the patient, and the required time for planning and placement, as well as the proficiency and availability of doctors. The process is tortuous, and the patient being in a severe or very critical condition makes stenting unsuitable. Previous case reports and studies have shown good results in terms of stent placement [10,24,25,26,27,28,29,30]. Therefore, we recommend stenting if the patient has underlying liver disease (Figure 7). If stenting is not suitable, hepatic artery sacrifice, rather than conservative treatment, is recommended [1,10]. A flow diagram for managing post-pancreatoduodenectomy hemorrhage is shown in Figure 7. 

However, it should be noted that there are several limitations to this study. First, this is a retrospective study, and there is inevitably bias in the selection of TAE types, due to the operator-dependent factor. Secondly, some subgroup analyses of patients according to TAE type have not been performed, due to the limited sample size. This small number of cases might cause a bias, even if this is one of the largest studies regarding TAE for post-pancreatoduodenectomy hemorrhage, to the best of our knowledge. Although the sample size is small, there are still some factors with a statistical difference, such as: the **rebleeding rate** between group B1 (15 patients) and group B-2 (4 patients) (0% vs 100%; *p* = 0.0003 < 0.05) and between sub-group B1 (15 cases) and group C (8 cases) (0% vs 62.5%; *p* = 0.0017 < 0.05); the **post-TAE complication rate** of group B **with/without underlying liver disease** (100% (3/3) vs. 23.1% (3/13); *p* = 0.036 < 0.05). All these differences are very significant and are not borderline. Finally, the patients included in the study received several different types of TAE, and it is difficult to attribute mortality to each type.

## 5. Conclusions

Post-DCP hemorrhage is uncommon and critical, and most surgeons do not have experience with this issue. Conservative treatment is acceptable if the initial angiographic findings are negative, but early hepatic artery sacrifice is suitable for patients with re-bleeding, even for those with negative results from the second angiography. Hepatic complications are not uncommon, particularly in patients with underlying hepatic disease, such as liver cirrhosis, and those who have had prior hepatic surgery. Conservative treatment, the embolization of the GDA stump, and incomplete hepatic artery sacrifice are not lasting solutions, as there is a higher risk of re-bleeding. Prophylactic hepatic artery sacrifice or stenting is recommended for patients experiencing postoperative bleeding. We hope our experience will be helpful to clinicians encountering such issues, in terms of their clinical care.

## Figures and Tables

**Figure 1 jpm-13-00264-f001:**
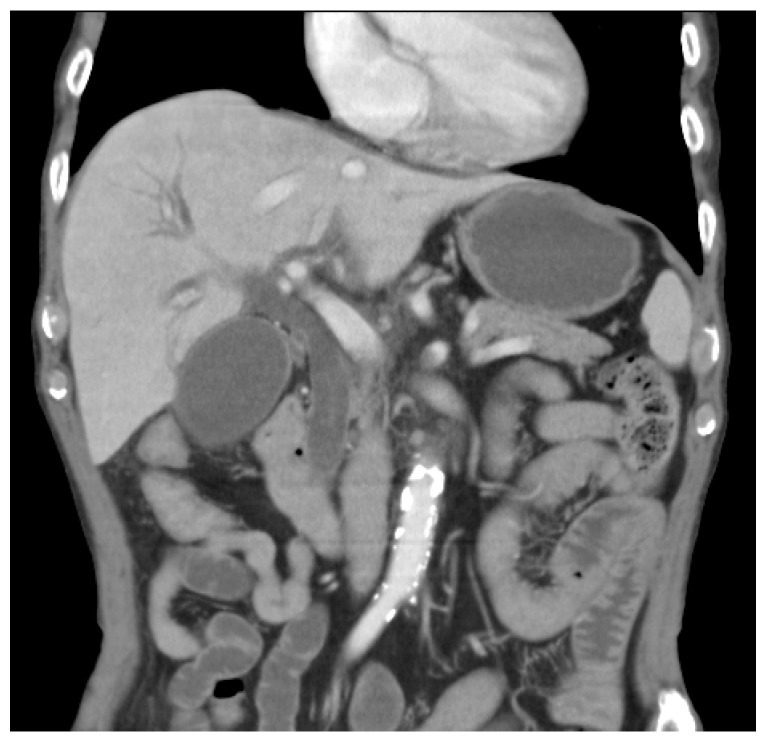
A 74-year-old male with a high-grade CBD obstruction due to a non-calcified nodule in the distal CBD orifice, which was highly indicative of bile-duct cancer. The pathology indicated intestinal metaplasia.

**Figure 2 jpm-13-00264-f002:**
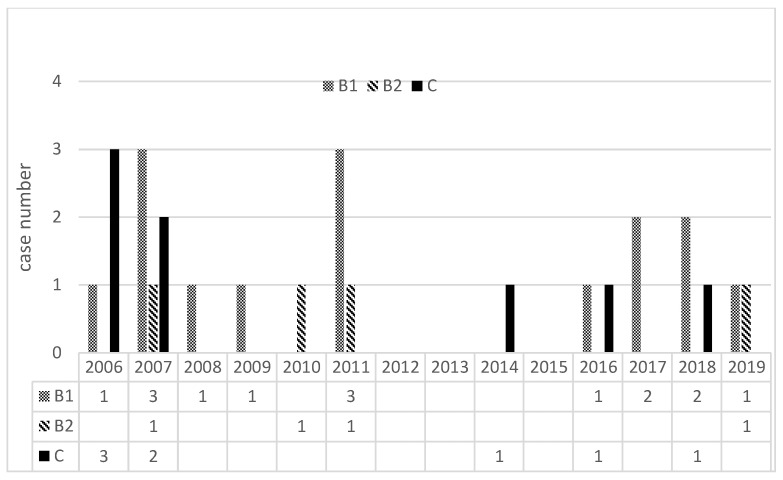
The number of cases for a specific treatment plan, shown according to the year in which the patient received the surgery and angiography.

**Figure 3 jpm-13-00264-f003:**
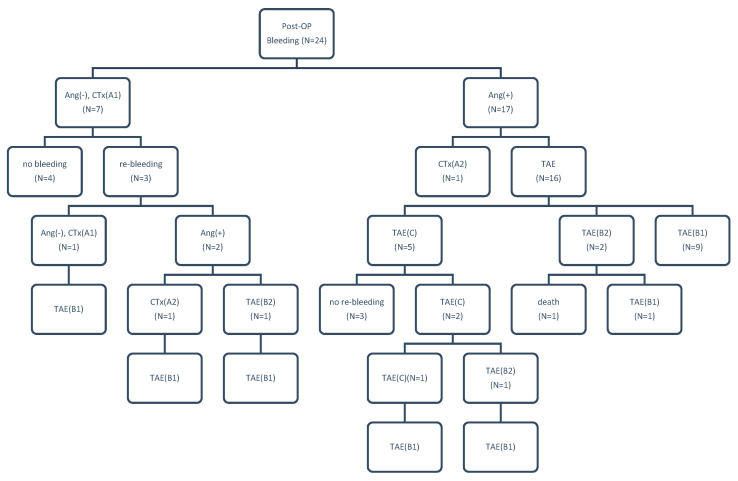
Flow chart representing 24 patients with postoperative hemorrhage after pancreatoduodenectomy (n = patient number). CTx, conservative treatment; Ang(+/−), positive or negative angiographic findings; B1, subgroup B1; B2, subgroup B2; C, group C.

**Figure 4 jpm-13-00264-f004:**
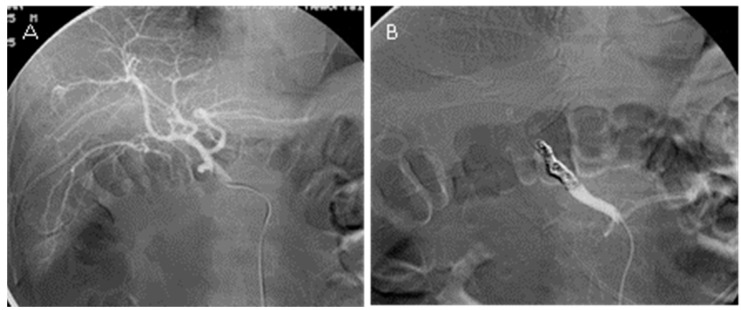
(**A**) Angiography image showing a pseudoaneurysm in the GDA stump without contrast extravasation. (**B**) Angiography after hepatic artery sacrifice, showing no opacify of the proper hepatic artery and GDA stump.

**Figure 5 jpm-13-00264-f005:**
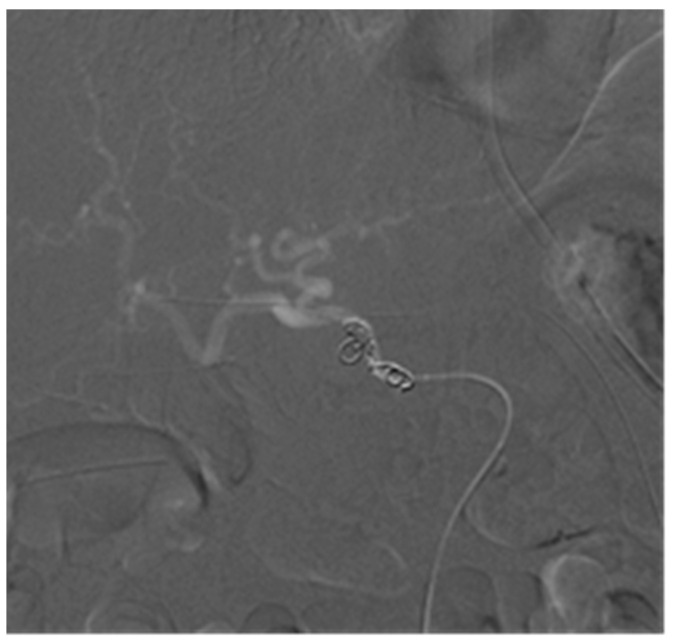
Angiography showing the preserved blood flow of the common and proper hepatic artery and distal branches after embolization with coils, but the GDA stump did not become filled with contrast material.

**Figure 6 jpm-13-00264-f006:**
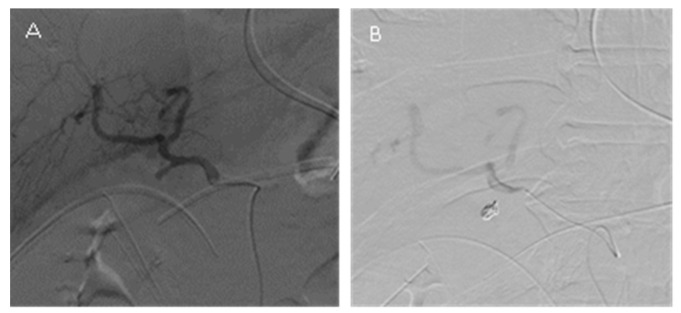
(**A**) Angiography showing a GDA stump pseudoaneurysm but there is no contrast extravasation. (**B**) After GDA stump embolization, angiography showed the total occlusion of the GDA stump.

**Figure 7 jpm-13-00264-f007:**
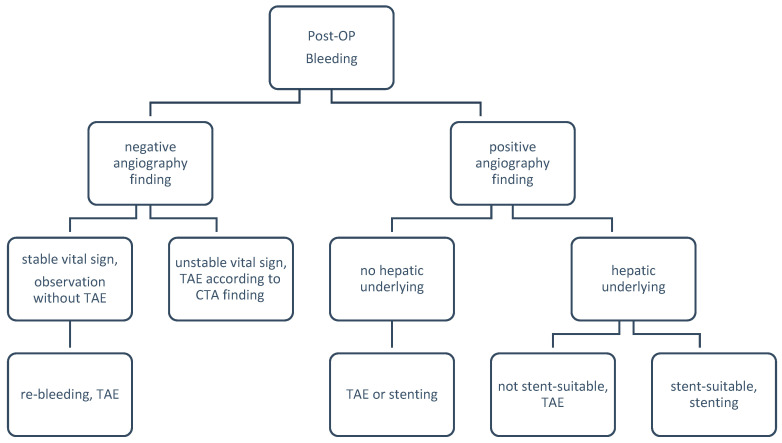
Flow diagram for the management of post-pancreatoduodenectomy hemorrhage.

## Data Availability

No open data are available.

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
