# Peer review of "Comparing the Clinical Efficacy of Coil Embolization in GDA Stump versus Common Hepatic Artery in Postoperative Hemorrhage after Pancreatoduodenectomy"

_jpm, 2023, doi:10.3390/jpm13020264_

Round 1
Reviewer 1 Report
thank you for the opportunity to review this multicentre retrospective series. the authors were interested in the interventional radiological therapeutic management of bleeding after pancreatectomy. of the 28 patients included, one patient did not have a pancreatectomy. Why was he included?
The patients mention an underlying liver disease. What is this? We are surprised that a patient with chronic liver disease could be eligible for CPD.
the authors describe 3 groups of patients A,B and C and subgroups in the results chapter but do not mention them in the material and methods chapter. why?
the authors report statistically significant differences. i'm not sure this makes statistical sense with so few patients given the very low number of events.
We do not know the number of patients who had a pancreatectomy during the long inclusion period (2004-2022). Did the authors take into account the changes in practice during the 18 years of inclusion in terms of diagnosis and therapeutic management?
Finally, this uni-centric retrospective study provides very little evidence on the therapeutic management of haemorrhage after pancreatectomy. It is also difficult to know the authors' main message from this study.
Reviewer 2 Report
The authors present an interesting work on postoperative haemorrhage after pancreatic surgery. The number of patients is ok and results from a rather large period - 18 years.
The findings are interesting, especially for surgeons and radiologists. The images are of good quality and the charts are very interesting.
Nevertheless, there are some aspects to improve.
The abstract should be improved to capture better the interest. There are at least two abbreviatures - TAE and GDA; they should appear in full. In line 13 the results star with the number of angiography and father the number of patients; usually the number of patients (study population is the first to be presented).
In M&M section, line 59, there is a case submitted to surgery due to intestinal metaplasia of the bile duct. This is a rather rare case and should be a little detailed. Was there dysplasia? high-risk features?
In the results section, line 105, the authors state the results by groups. However, the group formation is not described in the M&M section. It is stated only in the abstract: A(no embolization), B-1, etc... But even here there is no mention to the A1 and A2.... the group formation should be detailed in the methods with minimal descriptive statistics of all of them.
Diagrams have some abbreviations that should be explained in the caption: TAE; CTA, etc.
Round 2
Reviewer 1 Report
The manuscript has been considerably improved. however, there is still one patient without DCP included. the total number of patients who had PCD during the inclusion period is not known. The main limitation, of an observational study with few patients included, is to perform a statistical analysis that may induce biases given the very small number of tests. Finally, it would have been wise to discuss the placement of a covered stent graft instead of a permanent hepatic artery embolization.
